# Enantioselective synthesis of α-aminoboronates by NiH-catalysed asymmetric hydroamidation of alkenyl boronates

Yao Zhang [1], Deyong Qiao[1], Mei Duan[1], You Wang [1] ✉ & Shaolin Zhu [1,2] ✉

Chiral α-aminoboronic acids and their derivatives are generally useful as bioactive compounds and some have been approved as therapeutic agents. Here we report a NiH-catalysed asymmetric hydroamidation process that with a simple amino alcohol ligand can easily produce a wide range of highly enantioenriched α-aminoboronates from alkenyl boronates and dioxazolones under mild conditions. The reaction is proposed to proceed by an enantio-selective hydrometallation followed by an inner-sphere nitrenoid transfer and C−N bond forming sequence. The synthetic utility of this transformation was demonstrated by the efficient synthesis of a current pharmaceutical agent, Vaborbactam.

Enantioenriched α-aminoboronic acids and their derivatives are privileged structural elements commonly encountered in materials science and drug discovery (Fig. 1a)[1–3]. They have also been used as synthetically useful chiral building blocks in cross-coupling chemistry[4,5]. As a result, the development of catalytic synthetic methods to efficiently and selectively prepare molecules containing such high-value motifs from simple starting materials, has been the subject of intense research. In addition to traditional chiral auxiliary approaches[6–8], a number of catalytic asymmetric methods have been developed for the synthesis of α-aminoboronic acids, including borylation of aromatic aldimines[9–11], hydroboration of enamides[12,13] and hydrogenation[14–17] (Fig. 1b). As an alternative, a robust CuH-catalysed hydroamination[18–31] of easily available alkenylBdan (dan, naphthalene-1,8-diaminato) substrates was recently disclosed by Hirano and Miura[29]. Subsequently, a more efficient CuH-cascade hydroboration/hydroamination catalysis using readily available alkynes as starting materials was reported by Liu and Engle[31]. Despite the elegant nature of hydroamination reactions, a further four step sequence is needed to convert the obtained tertiary amines bearing an adjacent Bdan substituent to the desired α-aminoboronic acid pharmacophores and this has dramatically reduced the synthetic efficiency of the process. Consequently, the development of a complementary hydroamidation approach which could directly produce the pharmacophores - enantioenriched amides with an adjacent Bpin or B(OH)$_2$ group, is highly desirable.

Recently, our group and others have disclosed an asymmetric hydrofunctionalization platform that uses olefins directly as nucleo-philes and is enabled by a highly reactive chiral NiH catalyst[32–35]. This strategy is general and reliable and a large variety of electrophiles can be employed as coupling partners, which allows the stereochemically controlled formation of a variety of carbon–carbon[36–55] and carbon–heteroatom[56–65] bonds. Very recently, Seo and Chang[60,64], Yu[63], and our group[61] have demonstrated that dioxazolones are suitable electrophilic amidating reagents in NiH-catalysed reductive hydroamidation reactions. We envisioned that NiH-catalysed asymmetric hydrofunctionalization could be expanded to the asymmetric hydroamidation of alkenyl boronates with dioxazolones[30,60–64], thus enabling the direct synthesis of a variety of enantioenriched α-aminoboronates which have high potential in medicinal chemistry. As shown in Fig. 1c, with a structurally simple chiral amino alcohol ligand as a chiral source, an enantioenriched alkylnickel nucleophile would be formed through an enantiodifferentiating syn-hydronickellation reaction with an alkenylBpin substrate. Subsequent inner-sphere nitrenoid transfer with an amidating reagent[66–69] followed by a C−N bond formation would lead to the final chiral α-aminoboronate product.

In this work, we describe a highly enantioselective Ni-catalysed hydroamidation process enabled by a simple chiral amino alcohol ligand under exceptionally mild conditions. A wide variety of enantioenriched α-aminoboronates, a biologically active pharmacophore,

[1]State Key Laboratory of Coordination Chemistry, Jiangsu Key Laboratory of Advanced Organic Materials, Chemistry and Biomedicine Innovation Center (ChemBIC), School of Chemistry and Chemical Engineering, Nanjing University, Nanjing 210093, China. [2]School of Chemistry and Chemical Engineering, Henan Normal University, Xinxiang 453007, China. ✉e-mail: wangyou@nju.edu.cn; shaolinzhu@nju.edu.cn

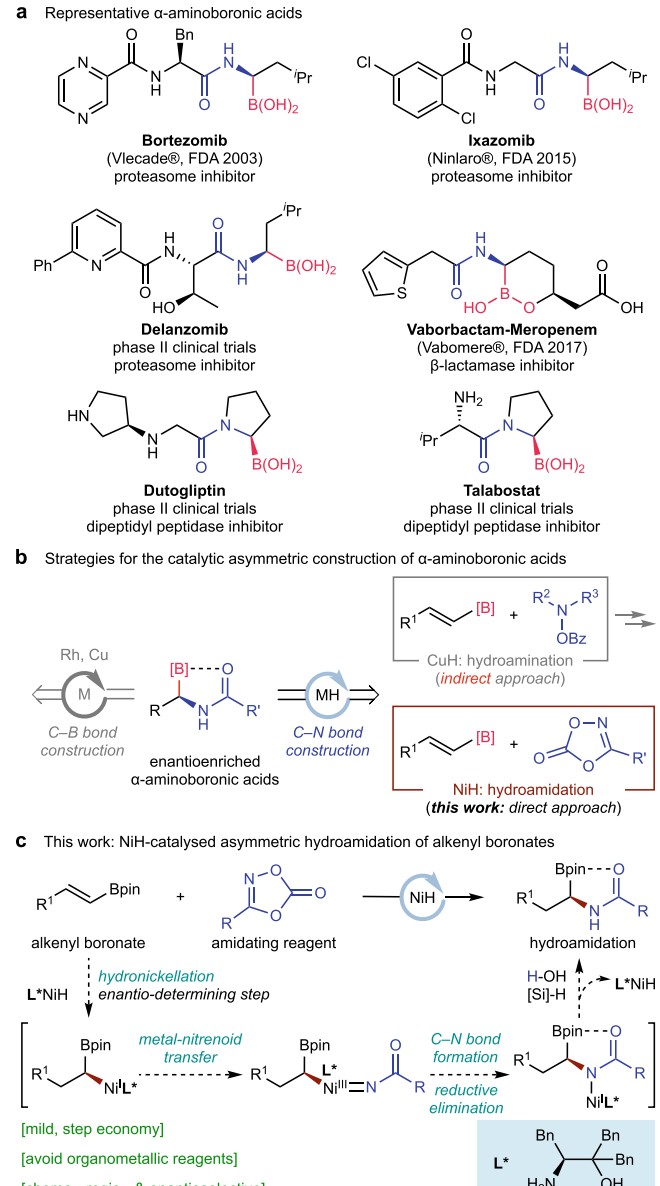

**Fig. 1 | Design plan: NiH-catalysed asymmetric hydroamidation to access bioactive chiral α-aminoboronic acids. a** Representative α-aminoboronic acids. **b** Strategies for the catalytic asymmetric construction of α-aminoboronic acids. **c** This work: NiH-catalysed asymmetric hydroamidation of alkenyl boronates. Bpin, pinacol boronic ester.

| entry | deviation from standard conditions | yield of 3a (%)* | ee (%)† |
|---|---|---|---|
| 1 | none | 75 (71) | 95 |
| 2 | NiCl₂·dme | 63 | 86 |
| 3 | **L1** instead of **L\*** | 67 | 93 |
| 4 | **L2** instead of **L\*** | 57 | 85 |
| 5 | **L3** instead of **L\*** | 14 | 13 |
| 6 | **L4** instead of **L\*** | 18 | 56 |
| 7 | **L5** instead of **L\*** | 74 | 44 |
| 8 | **L6** instead of **L\*** | 75 | 16 |
| 9 | (MeO)₂MeSiH instead of (EtO)₃SiH | 69 | 96 |
| 10 | *w/o* LiI | 46 | 97 |
| 11 | TBAI instead of LiI | 72 | 95 |
| 12 | *w/o* H₂O | 58 | 81 |
| 13 | MeOH instead of H₂O | 73 | 89 |
| 14 | NMP instead of DMA | 49 | 96 |
| 15 | (*Z*)-**1a** used | 68 (64) | 85 |
| 16 | under air in a closed vial | 68 | 94 |
| 17 | 2 equiv H₂O | 70 | 96 |

**Fig. 2 | Variation of reaction parameters.** *Yields were determined by gas chromatography (GC) using *n*-dodecane as the internal standard, the yield within parentheses is the isolated yield and is an average of two runs (0.20 mmol scale). †Enantioselectivities were determined by HPLC analysis. Bpin, pinacol boronic ester; DME, 1,2-dimethoxyethane; TBAI, tetrabutylammonium iodide; DMA, *N,N'*-dimethylacetamide; NMP, *N*-methyl-2-pyrrolidinone.

were directly obtained in high yields with excellent enantioselectivities. The utility of this protocol is illustrated by the synthesis of Vaborbactam in three steps.

## Results and discussions

### Reaction design and optimisation

Our initial studies focused on the enantioselective hydro-amidation of a pinacol-protected alkenylboronate (**1a**) using 3-phenyl-1,4,2-dioxazol-5-one (**2a**) as an amidating reagent (Fig. 2). We found that NiCl₂•6H₂O and the chiral amino alcohol ligand (**L\***) with triethoxysilane could afford the desired hydroamidation product (**3a**) in 71% isolated yield with 95% *ee* (entry 1). Other nickel sources such as NiCl₂•dme (dme = dimethoxyethane) led to lower yields and *ee* (entry 2). Other ligands (**L1**–**L6**) also gave significantly lower yields and *ee* (entries 3–8). Dimethoxy(methyl) silane was shown to be a less effective silane (entry 9). Addition of a

catalytic amount of an iodide salt was found to improve the yield (entry 10) and LiI was proved to be the best additive (entry 11). Currently, the exact role of LiI is still under investigation. Inclusion of an extra proton source improved both the yield and the *ee* (entry 12), but alcohol was less effective than H₂O (entry 13). Through DFT calculation, Chang, Seo and coworkers have demonstrated that the transmetalation between Ni-enamido complex and hydrosilane is thermodynamically unfavorable, and the addition of H₂O could provide thermodynamic driving force for this step by an irreversible Si–O bond formation[60]. Similarly, with other solvents, the reaction proceeded less efficiently (entry 14). Notably, the *E,Z*-configuration of alkenyl boronates has a significant effect on the *ee* of the products. For example, a diminished *ee* was observed when (*Z*)-**1a** was used (entry 15). Notably, the reaction is insensitive to air (entry 16) and moisture (entry 17).

### Substrate scope

Under the optimal conditions, the scope of the alkenyl boronate partner is fairly broad (Fig. 3). Substrates containing a variety of functional groups, including an alkyl chloride (**3f**, **3g**), a variety of ethers (**3h**, **3i**, **3l**, **3o** and **3p**), esters (**3j**, **3l**–**3p**), as well as sulfonamides (**3k**, **3m**), were shown to be competent. Notably, the reaction is orthogonal to alkyl chlorides (**3f**, **3g**), a potential coupling handle for further derivatization. The generality of this protocol was further highlighted by the successful introduction of several core structures of bioactive

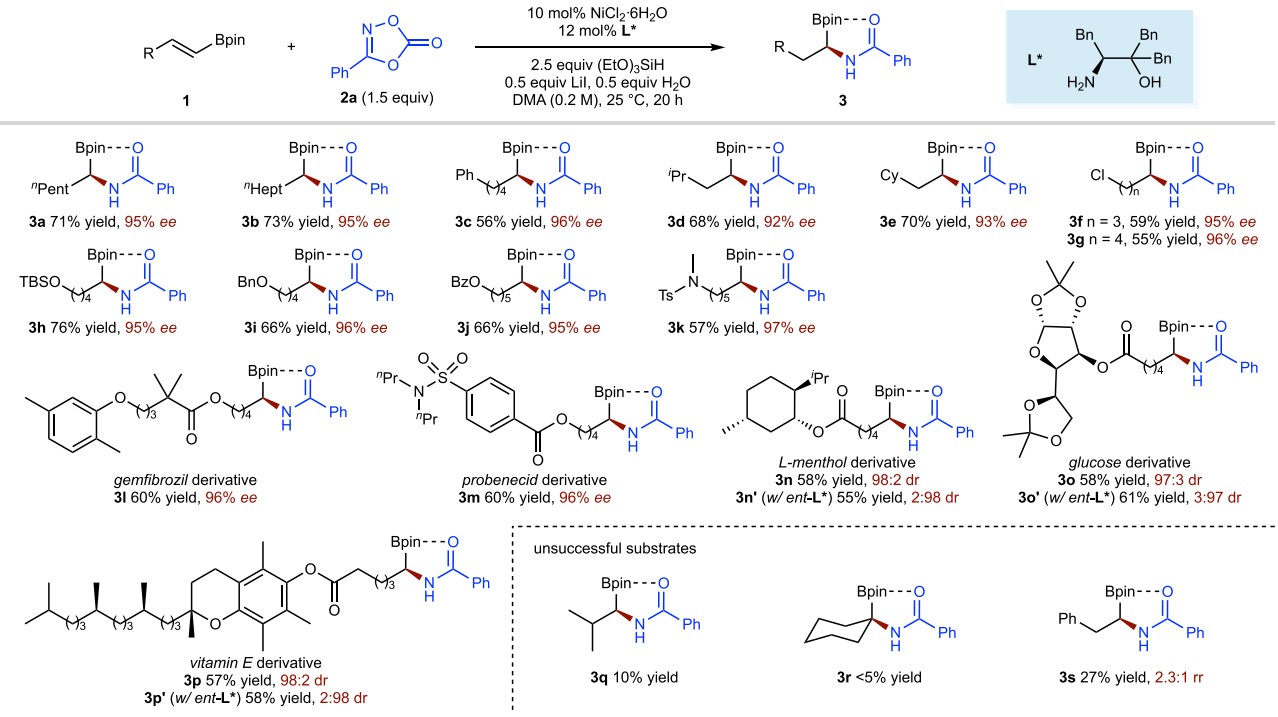

**Fig. 3 | Substrate scope of alkenyl boronate coupling partner.** Yield under each product refers to the isolated yield of purified product (0.20 mmol scale, average of two runs), enantioselectivities were determined by chiral HPLC analysis. Bpin, pinacol boronic ester; *n*Hept, *n*-heptyl.

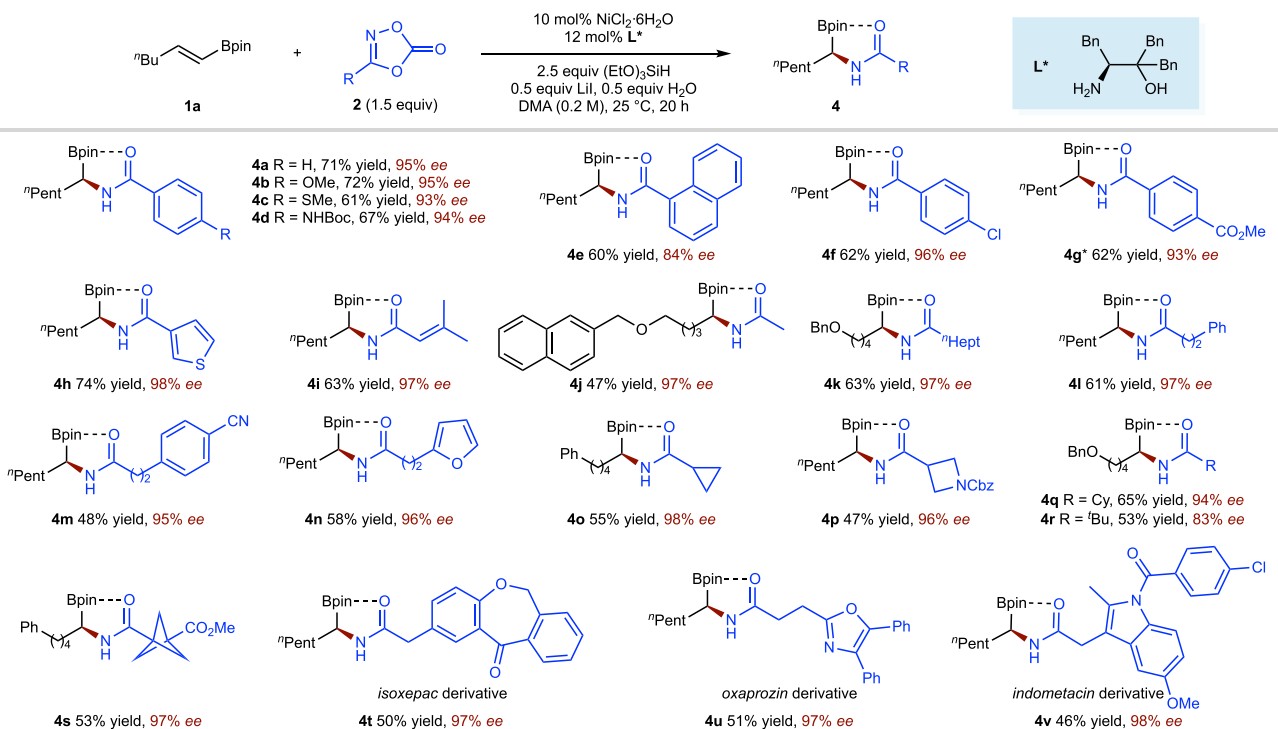

**Fig. 4 | Substrate scope of alkene coupling component.** Under each product is given yield and enantioselectivities (*ee*) in percent. Yield and *ee* are as defined in Fig. 3 legend. *MeOH instead of H$_2$O.

and pharmaceutical molecules, including gemfibrozil (**3l**), probenecid (**3m**), menthol (**3n**), glucose (**3o**), and vitamin E (**3p**), which afforded molecules with two bioactive fragments that are useful in pharmaceutical research. High levels of diastereoselectivity were achieved in the hydroamidation of substrates (**3n**, **3n'**, **3o**, **3o'**, **3p**, **3p'**) derived from a series of chiral molecules. Unfortunately, β,β-disubstituted and α,β-disubstituted alkenyl boronates, which are less reactive towards

hydronickellation, produced the desired products (**3q**, **3r**) with low yields under these conditions. When β-aryl substituted alkenyl boronate (**3s**) was used, amidation happened at both the α-carbon atom of the boronate and the benzylic position as a 2.3:1 mixture.

A subsequent survey of possible dioxazolone components revealed a wide range of aromatic (**4a–4h**), alkenyl (**4i**) and aliphatic (**4j–4v**) amide electrophiles as competent substrates (Fig. 4)[66–69]. For

**Fig. 5 | Gram-scale experiment and synthetic application. a** Gram-scale experiment. **b** One-pot asymmetric hydroamidation w/o isolation of alkenyl boronate. **c** Concise synthetic route to Vaborbactam.

aromatic amide electrophiles, both electron-donating (**4b**–**4d**) and electron-withdrawing (**4f**, **4g**) substituents on the benzene ring of the electrophile, as well as a heterocyclic electrophile (**4h**) underwent asymmetric hydroamidation smoothly. For alkyl amide electrophiles, a various primary (**4j**–**4n**, **4t**–**4v**), secondary (**4o**–**4q**), and tertiary (**4r**, **4s**) aliphatic amide electrophiles all were efficiently converted into the corresponding amide products. Functional groups, including a variety of ethers (**4b**, **4c**, **4j**, **4k**, **4q**, **4r**, **4t**, **4v**), carbamates (**4d**, **4p**), aryl chlorides (**4f**, **4v**), esters (**4g**, **4s**), a nitrile (**4m**) and an easily reduced ketone (**4t**), were left intact. Heterocycles such as thiophene (**4h**), furan (**4n**), oxazole (**4u**), and indole (**4v**) were also accommodated. Successful functionalization of a series of biologically important compounds such as isoxepac (**4t**), oxaprozin (**4u**), and indomethacin (**4v**) was achieved, demonstrating the potential utility of this protocol in the late-stage functionalization of complex molecules.

## Application

The preparative utility of this process was highlighted by a 5 mmol scale experiment (Fig. 5a). Product **3a** was obtained without notable erosion of the yield or the enantioselectivity (cf. Fig. 2, entry 1, 71% yield, 95% *ee*). As shown in Fig. 5b, the desired hydroamidation product (**3a**) could also be obtained directly from alkyne through a one-pot reaction sequence without isolating the hydroboration intermediate (**1a**). The synthetic utility was further demonstrated by a three-step synthesis of Vaborbactam, a β-lactamase inhibitor (Fig. 5c)[70]. The key hydroamidation of the alkenyl boronate (**6**), obtained through Zr-

catalysed hydroboration, generated the α-aminoboronate product (**7**) in moderate yield (50%) with high diastereoselectivity (97:3 dr). The synthesis of target compound, Vaborbactam was completed by HCl-mediated deprotection.

## Mechanistic investigation

A series of experiments were carried out to gain insight into the reaction mechanism. A linear correlation between the *ee* value of the ligand **L*** and that of the product **3a** was observed (Fig. 6a), an observation that is consistent with the monomeric nature of the active catalyst. Since $H_2O^{38}$ and hydrosilane could both act as a hydride source, an isotopic labelling experiment was carried out using $D_2O$ (Fig. 6b, top). No deuterium incorporation was observed in the product (**3a**), eliminating the possibility of a protic reagent as the hydride source. To gain insight into the hydrometallation process, the reaction of a deuterium labeled olefin (**1b-D**) was evaluated (Fig. 6b, bottom). Diastereomerically pure **3b-D** was obtained from this reaction, indicating that *syn*-hydronickellation is involved in the enantio-determining step. This conclusion is also consistent with the observation that the *E,Z*-configuration of alkenyl boronates has a significant effect on the ee of the products (cf. Fig. 2, entry 1 vs. entry 15).

To further understand the subsequent amidation process, we treated dioxazolone (**2a**) with triphenylphosphine (PPh₃) under standard conditions (Fig. 6c)[60]. A nitrene transfer to the phosphine adduct, imidophosphorane was obtained, suggesting that the formation of a

**a** Nonlinear effect

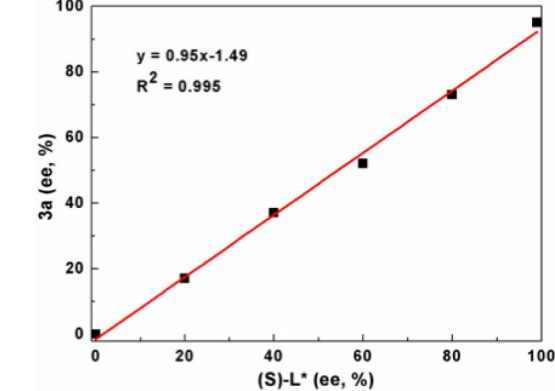

**b** Isotopic labelling experiments

NiCl$_2$·dme (10 mol%)
**L*** (12 mol%)

(EtO)$_3$SiH (2.5 equiv)
LiI (0.5 equiv)
D$_2$O (2.0 equiv )
DMA (0.2 M), 25 °C, 20 h

**1a**   **2a** (1.5 equiv)   (no D was detected)
**3a** 57% yield, 96% *ee*

protic reagent is **not the hydride source**

Std conditions

**1b-D** (93% D)   **2a** (1.5 equiv)   (0.93 D) (0 D)
**3b-D** 70% yield, 96% *ee*

*syn*-hydronickellation is proposed to be the **enantio-determining step**

**c** Capture of metal-nitrenoid intermediate

NiCl$_2$·6H$_2$O (10 mol%)
**L*** (12 mol%)

(EtO)$_3$SiH (2.5 equiv)
LiI (0.5 equiv), H$_2$O (0.5 equiv)
DMA (0.2 M), 25 °C, 20 h

'Std conditions'

**2a**   (2.0 equiv)   81% yield
*w/o* Ni: <5% yield

**Fig. 6 | Preliminary mechanistic experiments. a** Nonlinear effect. **b** Isotopic labelling experiments. **c** Capture of metal-nitrenoid intermediate.

nickel-nitrenoid species could be one possible pathway for amidation process.

In conclusion, we are reporting development of a NiH-catalysed enantioselective hydroamidation procedure which enables the facile synthesis of a variety of enantioenriched α-aminoboronates with high potential in medicinal chemistry. With a simple chiral amino alcohol ligand as the source of chirality, a broad range of both alkenyl boronates and dioxazolone partners are suitable for this transformation. This mild and straightforward hydroamidation process has been applied to the efficient synthesis of a pharmaceutical agent.

## Methods

### General procedure (A) for NiH-catalysed asymmetric hydroamidation of alkenyl boronates

In a nitrogen-filled glove box, to an oven-dried 8 mL screw-cap vial equipped with a magnetic stir bar was added NiCl$_2$·6H$_2$O (4.8 mg, 10 mol%), **L*** (8.0 mg, 12 mol%), LiI (13.4 mg, 0.10 mmol, 0.50 equiv), 1,4,2-dioxazol-5-one (0.30 mmol, 1.5 equiv) (if the olefin is a solid, it was also added at this time), and anhydrous DMA (1.0 mL, 0.20 M). The mixture was stirred for 10 min at rt, at which time alkenyl boronate (0.20 mmol, 1.0 equiv) (if the 1,4,2-dioxazol-5-one is a liquid, it was added at this time), H$_2$O (1.8 μL, 0.10 mmol, 0.50 equiv), and (EtO)$_3$SiH

(92 μL, 0.50 mmol, 2.5 equiv) were added to the resulting mixture in this order. The tube was sealed with a teflon-lined screw cap, removed from the glove box and the reaction was stirred at 25 °C water bath for up to 20 h (the mixture was stirred at 800 rpm). After the reaction was complete, the reaction was quenched upon the addition of H$_2$O, and the mixture was extracted with Et$_2$O. The organic layer was concentrated to give the crude product. *n*-Dodecane (20 μL) was added as an internal standard for GC analysis. The product was purified by flash column chromatography (petroleum ether/EtOAc) for each substrate. The yields reported are the average of at least two experiments, unless otherwise indicated. The enantiomeric excesses (% *ee*) were determined by HPLC analysis using chiral stationary phases.

## Data availability

The authors declare that the main data supporting the findings of this study, including experimental procedures and compound characterization, are available within the article and its Supplementary information files, and also are available from the corresponding authors.

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

## Acknowledgements

Support was provided by NSFC (92156004, 22271143, 22271146), NSF of Jiangsu Province (BK20190281, BK20201245), programs for high-level entrepreneurial and innovative talents introduction of Jiangsu Province (group program), Fundamental Research Funds for the Central Universities (020514380282), and Open Research Fund of School of Chemistry and Chemical Engineering, Henan Normal University.

## Author contributions

S.Z. and Y.W. designed and supervised the project. Y.Z., D.Q., and M.D. performed and analysed the experiments. All authors co-wrote the manuscript, analysed the data, discussed the results, commented on the manuscript, and approved the final version of the manuscript.

## Competing interests

The authors declare the following competing interest(s): S.Z. and Y.Z. are inventors on a patent application number CN202210526440.5 which is based on the synthesis of Vaborbactam using this method. D.Q., M.D., and Y.W. declare no competing interests.
