## [Peer Review File · Nature Communications]

REVIEWER COMMENTS

Reviewer #1 (Remarks to the Author):

Zhu and co-authors propose here a manuscript entitled α -Aminoboronates by NiH-catalysed Asymmetric Hydroamidation of Alkenyl Boronates, for consideration in the journal of Nature Communications. This is a very interesting and useful reaction, that enables the hydroamidation of alkenyl boronates, which are useful synthons, with dioxazolones. While the yields are moderate to good, the enantioselectivities of products obtained are excellent, providing an efficient method for the synthesis of enantioenriched. The chiral amino alcohol supported Ni catalyst was specially optimized to enable this reaction. The author also carried out several control experiments to investigate the reaction mechanism. The article is otherwise clearly written, and the SI is of high quality. I therefore recommend acceptance. My only comments before acceptance concern:

In the part of reaction optimization, water was addressed to affect both the yield and the enantioselectivity of the product (Figure 2, entries 12). A discussion of the role of water in this hydroamidation would help readers for a better understanding. Also, what's the effect of LiI?

For the scope of alkenyl boronates in Figure 3, the reaction conditions shown a high compatibility of linear substrates; how about alkenyl boronates bearing a branched substituent?

In the part of mechanistic investigation, a relevant literature for the metal-nitrenoid intermediate trapping experiment (J. Am. Chem. Soc. 2019, 141, 19048) should be cited.

Reviewer #2 (Remarks to the Author):

Recommendation: publish in Nature Communications with minor revisions.

Comments:

Developing catalytic asymmetric synthesis methods to make enantioenriched α -aminoboronate is drawing intensive attention in recent organic chemistry and drug discovery. Zhu and coworkers

described a highly enantioselective hydroamidation reaction of alkenyl boronates with dioxazolones by Ni-H catalysts. A very nice aspect of this manuscript is that the simple and easily available amino alcohols are proved to be efficient ligands for Ni-H catalyzed asymmetric hydrofunctionalization reaction. For the substrate scope, both alkenyl boronates and amide electrophiles (aromatic, alkenyl, and aliphatic dioxazolones) are well tolerated in this catalytic system. Synthetic transformation of the drug molecular and gram-level experiment further demonstrates the useful and practical method. The manuscript is well-written and the conclusions are sufficiently supported by the experimental results. The SI of this manuscript is complete and presented in good shape. Therefore, I would recommend acceptance of this nice work in Nature Communication after the following minor revisions are addressed:

1. The title of this manuscript is not clear. I recommend using “Enantioselective synthesis of α -aminoboronates by NiH-catalysed asymmetric hydroamidation of alkenyl boronates” as the title of this manuscript.
2. The results of aryl substituted alkene substrates should be provided.
3. Please change the “catalyzed” to “catalysed” with the unified style?

Reviewer #3 (Remarks to the Author):

In this manuscript, Zhu and co-workers report the synthesis of chiral α -aminoboronic acids using Ni-hydride catalyzed asymmetric hydroamidation. Compared to hydrogenation precedents that synthesize enantioenriched α -aminoboronic acids, this method circumvents the use of expensive rhodium and iridium catalysts. This report also addresses limitations associated with CuH-catalyzed methods. The authors report a robust method that can handle a wide range of alkenyl boronates as well as amide electrophiles including biologically relevant handles. The manuscript is suitable for publication in Nat. Comm., provided the authors can address following comments:

1. Reference 14 seems irrelevant to synthesizing α -aminoboronic acids.
2. Deuterium labeling study, can the authors use (EtO)₃Si-D?
3. In page 51 in the SI where the authors show the yield of 3a vs time, the yield written on the table does not match the yield shown in the graph. Entry 3 shows that at hour 4 the yield is 73 but it is clearly not 73 in the graph.
4. SI generally looks good. Certain HPLC traces contain slight impurities that the author's don't account for (i.e. 4l, 3g, 3j, 3k, 4c)

In response to **reviewer 1** (quotes from reviewer are italicized):

Reviewer #1 (Remarks to the Author):

Zhu and co-authors propose here a manuscript entitled α -Aminoboronates by NiH-catalysed Asymmetric Hydroamidation of Alkenyl Boronates, for consideration in the journal of Nature Communications. This is a very interesting and useful reaction, that enables the hydroamidation of alkenyl boronates, which are useful synthons, with dioxazolones. While the yields are moderate to good, the enantioselectivities of products obtained are excellent, providing an efficient method for the synthesis of enantioenriched. The chiral amino alcohol supported Ni catalyst was specially optimized to enable this reaction. The author also carried out several control experiments to investigate the reaction mechanism. The article is otherwise clearly written, and the SI is of high quality. I therefore recommend acceptance. My only comments before acceptance concern:

1. In the part of reaction optimization, water was addressed to affect both the yield and the enantioselectivity of the product (Figure 2, entries 12). A discussion of the role of water in this hydroamidation would help readers for a better understanding. Also, what's the effect of Lil?

Through DFT calculation (Ref 60), Chang, Seo and coworkers have demonstrated that the transmetalation between Ni-enamido complex and hydrosilane is thermodynamically unfavorable, and the addition of H₂O could provide thermodynamic driving force for this step by an irreversible Si–O bond formation. We have now added this information in the main text.

As shown in Fig. 2, addition of a catalytic amount of an iodide salt was found to improve the yield (entry 10) and Lil was proved to be the best additive (entry 11). Currently, the exact role of Lil is still under investigation. We have now also added this information in the main text.

2. For the scope of alkenyl boronates in Figure 3, the reaction conditions shown a high compatibility of linear substrates; how about alkenyl boronates bearing a branched substituent?

As shown in the modified main text and Figure 3, we have now added this information, “Unfortunately, β,β -disubstituted and α,β -disubstituted alkenyl boronates, which are less reactive towards hydronickellation, produced the desired products (**3q**, **3r**) with low yields under these conditions.”.

3. In the part of mechanistic investigation, a relevant literature for the metal-nitrenoid intermediate trapping experiment (J. Am. Chem. Soc. 2019, 141, 19048) should be cited.

We have now cited the above-mentioned paper (see: ref. 68). We apologize for missing the above-mentioned paper.

In response to **reviewer 2** (quotes from reviewer are italicized):

Reviewer #2 (Remarks to the Author):

Developing catalytic asymmetric synthesis methods to make enantioenriched α -aminoboronate is drawing intensive attention in recent organic chemistry and drug discovery. Zhu and coworkers described a highly enantioselective hydroamidation reaction of alkenyl boronates with dioxazolones by Ni-H catalysts. A very nice aspect of this manuscript is that the simple and easily available amino alcohols are proved to be efficient ligands for Ni-H catalyzed asymmetric hydrofunctionalization reaction. For the substrate scope, both alkenyl boronates and amide electrophiles (aromatic, alkenyl, and aliphatic dioxazolones) are well tolerated in this catalytic system. Synthetic transformation of the drug molecular and gram-level experiment further demonstrates the useful and practical method. The manuscript is well-written and the conclusions are sufficiently supported by the experimental results. The SI of this manuscript is complete and presented in good shape. Therefore, I would recommend acceptance of this nice work in Nature Communication after the following minor revisions are addressed:

1. *The title of this manuscript is not clear. I recommend using “Enantioselective synthesis of α -aminoboronates by NiH-catalysed asymmetric hydroamidation of alkenyl boronates” as the title of this manuscript.*

We have now changed the title accordingly.

2. *The results of aryl substituted alkene substrates should be provided.*

When aryl substituted alkene substrates were used, the hydroamidation products were obtained with low regioselectivities and yields under current conditions. We have now added one such example in Fig. 3. We have also added a sentence in the Main text, “When β -aryl substituted alkenyl boronate (**3s**) was used, amidation happened at both the α -carbon atom of the boronate and the benzylic position as a 2.3:1 mixture.”.

In addition, two extra successful substrates (**3o** and **3p**) were added into Fig. 3.

3. *Please change the “catalyzed” to “catalysed” with the unified style?*

We have now changed the “catalyzed” to “catalysed” and fixed several typos.

In response to **reviewer 3** (quotes from reviewer are italicized):

Reviewer #3 (Remarks to the Author):

In this manuscript, Zhu and co-workers report the synthesis of chiral α -aminoboronic acids using Ni-hydride catalyzed asymmetric hydroamidation. Compared to hydrogenation precedents that synthesize enantioenriched α -aminoboronic acids, this method circumvents the use of expensive rhodium and iridium catalysts. This report also addresses limitations associated with CuH-catalyzed methods. The authors report a robust method that can handle a wide range of alkenyl boronates as well as amide electrophiles including biologically relevant handles. The manuscript is suitable for publication in Nat. Comm., provided the authors can address following comments:

1. *Reference 14 seems irrelevant to synthesizing α -aminoboronic acids.*

We have now deleted this reference.

2. *Deuterium labeling study, can the authors use (EtO)₃Si-D?*

We sincerely thank the reviewer for this suggestion. However, this deuterated silane is not commercial available and hard to synthesize in lab (for the consideration of potential hazard). Meanwhile, the HBpin/DBpin is not a suitable hydride/deuterium source for the current reaction. So we used deuterium labeled alkenyl boronate to carry out the isotopic labelling experiment instead.

3. *In page 51 in the SI where the authors show the yield of 3a vs time, the yield written on the table does not match the yield shown in the graph. Entry 3 shows that at hour 4 the yield is 73 but it is clearly not 73 in the graph.*

We apologize for this typo. At hour 4 the yield is 59%. We have now double-checked and revised the data accordingly.

4. *SI generally looks good. Certain HPLC traces contain slight impurities that the author’s don’t account for (i.e. 4l, 3g, 3j, 3k, 4c).*

For compounds **3g**, **3j**, **3k**, **4c** and **4l** mentioned above, we have now re-run the reactions, re-purified the product and re-run the corresponding HPLC traces. In some cases, the slight impurity (peak appeared at around 3.5-4.0 min in HPLC trace) might be trace amount ethyl acetate, now the HPLC traces are clean.

REVIEWERS' COMMENTS

Reviewer #1 (Remarks to the Author):

I do not have any further concern, and recommend acceptance.

Reviewer #2 (Remarks to the Author):

In the revised manuscript, the authors addressed all the concerns appropriately and made the necessary revisions. This reviewer recommends publishing this paper in Nature Communication as it stands.